# Medical face masks offer self-protection against aerosols: An evaluation using a practical *in vitro* approach on a dummy head

Christian M. Sterr, Inga-Lena Nickel, Christina Stranzinger, Claudia I. Nonnenmacher-Winter, Frank Günther 📷 *

Division of Infection Control and Hospital Epidemiology, Philipps University Marburg, Marburg, Germany

* frank.guenther@staff.uni-marburg.de

**Data Availability Statement:** All relevant data are within the manuscript and its Supporting Information files.

## Abstract

Since the appearance of severe acute respiratory syndrome coronavirus 2 (SARS-CoV-2), the question regarding the efficacy of various hygiene measures and the use of personal protective equipment (PPE) has become the focus of scientific and above all public discussion. To compare respirators, medical face masks, and cloth masks and determine if it is recommendable to wear face masks to protect the individual wearer of the mask from inhaling airborne particles, we challenged 29 different masks with aerosols and tested the pressure drop as a surrogate for breathing resistance owing to the mask material. We found that Type II medical face masks showed the lowest pressure drop ($12.9\pm6.8$ Pa/cm$^2$) and therefore additional breathing resistance, whereas respirators such as the KN95 ($32.3\pm7.0$ Pa/cm$^2$) and FFP2 ($26.8\pm7.4$ Pa/cm$^2$) showed the highest pressure drops among the tested masks. The filtration efficacy of the mask material was the lowest for cloth masks ($28\pm25\%$) followed by non-certified face masks ($63\pm19\%$) and certified medical face masks ($70\pm10\%$). The materials of the different respirators showed very high aerosol retentions (KN95 [$94\pm4\%$] and FFP2 [$98\pm1\%$]). For evaluating the as-worn filtration performance simulating real live conditions each mask type was also tested on a standardized dummy head. Cloth masks and non-EN-certified face masks had the worst as-worn filtration efficacies among the tested masks, filtering less than 20% of the test aerosol. Remarkably, certified type II medical face masks showed similar (p>0.5) as-worn filtration results ($47\pm20\%$) than KN95 masks ($41\pm4\%$) and FFP2 masks ($65\pm27\%$), despite having a lower pressure drop. Face shields did not show any significant retention function against aerosols in our experiment. Our results indicate that it seems recommendable to wear face masks for providing base protection and risk reduction against inhaling airborne particles, in low-risk situations. In our study, especially EN 14683 type II certified medical face masks showed protective effectiveness against aerosols accompanied by minimal additional breathing resistance. FFP2 Respirators, on the other hand, could be useful in high-risk situations but require greater breathing effort and therefore physical stress for users.

**Funding:** The authors received no specific funding for this work.

**Competing interests:** The authors have declared that no competing interests exist.

# Background/Introduction

Severe acute respiratory syndrome coronavirus 2 (SARS-CoV-2) is one of seven coronaviruses known to elicit infections in humans and is very similar to the SARS virus of 2002 [1,2]. From what is known to date, its transmission mainly occurs via droplets (particles >5 μm) and aerosols (particles <5 μm) in poorly ventilated settings [3,4]. Recent findings suggest that SARS-CoV-2 can be transmitted before the symptoms occur [5].

Besides social distancing and quarantine, one of the major factors preventing the spread of the virus is the practice of face covering [6]. Nurses and doctors are unable to maintain distances from their patients and are at high risks of acquiring the infection and transmitting it to others [7]. Furthermore, in a pandemic situation, their workforce is the most essential [8]. Therefore, it is crucial to provide health care workers (HCWs) with high-quality face masks or respirators to offer protection to them and their patients [8]. In the medical field, surgical or medical face masks can be differentiated from FFP or N95 respirators [9,10]. All masks and respirators are usually tested against the European standards (EN) to ensure reliable quality [11,12]. EN 149 is used for testing respirators, and EN 14683 is used for testing medical face masks [11,12]. Respirators are usually recommended for preventing airborne diseases such as tuberculosis or measles, although they have not been tested for the passage of bioaerosols according to the EN 149 standard [10,11]. The use of medical face masks is recommended for protection from infections caused by droplets [10]. At the beginning of the SARS-CoV-2 outbreak, some experts, especially in Europe, advised against the usage of medical face masks for protection against SARS-CoV-2 [6]. This might be due to the testing method of medical face masks according to the EN 14683 standard. In that norm, only the material is tested without considering the mask fit [12]. Moreover, these tests only assess third-party protection without considering self-protection properties [12]. However, these recommendations should be questioned critically. Loeb et al. showed that the protective effects of medical face masks and respirators against influenza are not significantly different [9].

In a non-pandemic situation, it is easy to understand the recommendations to wear masks, given the supply of certified medical face masks and respirators. However, the SARS-CoV-2 outbreak has led to a shortage of personal protective equipment (PPE) such as medical face masks and respirators. Moreover, many newly produced masks seemed to be of poor quality. Soon after, there were warnings of fake masks in the global market [13]. Particularly, masks labeled as KN95 were notified to the Rapid Alert System for Dangerous Non-food Products (RAPEX), a black list of the European Union. In addition, do-it-yourself (DIY) or cloth masks produced without certificates and of variable quality were available everywhere. Because of the crucial role that HCWs play and the uncertainty of the protective function of many newly offered masks, we decided to test every mask type intended for HCWs before using them in hospital. To compare respirators, medical face masks, and cloth masks and determine if it is recommendable to wear face masks to protect the individual wearer of the mask from inhaling airborne particles, we challenged 29 different masks and 3 face shields with di-ethyl-hexyl-sebacat (DEHS) aerosols and tested pressure drops over the mask materials.

# Methods

## Test system

We conducted three different experiments to compare the filtration efficacies of 32 different respirators, surgical face masks, cloth masks and face shields. All the mask and respirator models were tested using three representative specimens to rule out material defects; for some cloth masks however, only two specimens were available. First, we assessed the material properties

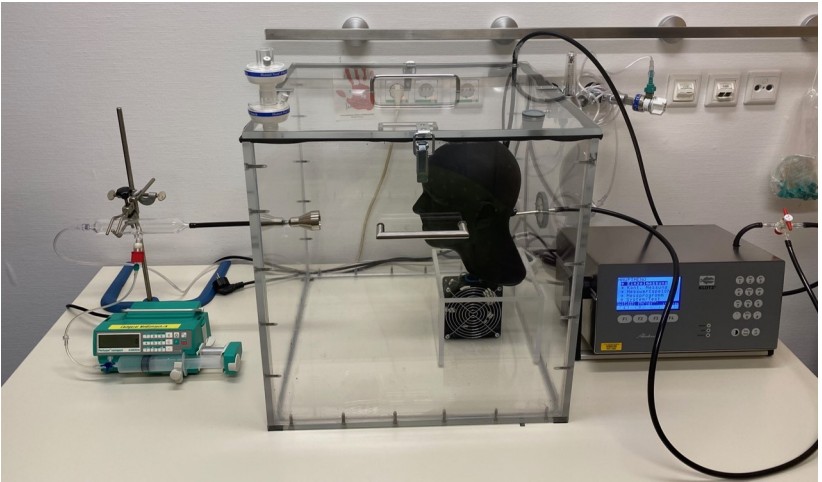

**Fig 1. Picture of the practical mask test system with an aerosol applicator, a particle-tight, closed Plexiglas chamber with standardized test head, and a connected particle counter.**

of the masks, including filtration efficacy and pressure difference, without considering mask fit. To assess the as-worn filtration efficacies, including mask fit, the masks were mounted on a standard-sized dummy head, which was 3D printed according to the measurements from Zhuang et Bradtmiller to correspond to the average head size of the U.S. population [14]. An air collector with an opening width of 28 mm was built in an airtight fashion into the dummy head as an artificial airway. To imitate a skin-like surface for a more realistic mask fit, the dummy head was rubber coated with liquid gum (Peter Kwasny, Gundelsheim, Germany) after 3D printing. The dummy head was placed into an airproof acrylic glass chamber with an edge length of 0.5 m (Fig 1). To test the material properties of the masks, we used a single standard-sized air collector with an opening width of 40 mm and fixed the masks in an airtight fashion on the collector. The air collector was then placed in the chamber instead of the dummy head.

We used DEHS (Sigma Aldrich, Germany) aerosols produced by an ATM 225 E aerosol generator (Markus Klotz, Bad Liebenzell, Germany). DEHS produces very stable aerosols and is therefore well suited for our experiments. Before starting the experiment, the chamber was flooded with particle-free medically pressurized air.

## Self-protection and material filtration efficacy

To test the material filtration properties with the standard air collector and assess the as-worn self-protection filtration efficacy, particle-free pressurized air was mixed with the produced aerosol in a cylindrical glass flask that was connected to the chamber approximately 20 cm in front of the air collector or dummy head. The aerosol was released continuously from a glass needle into the glass flask with a syringe pump to control the amount and maintain a homogeneous concentration of particles inside the chamber. The particle counter Abakus Air (Markus Klotz, Bad Liebenzell, Germany) was connected either to the artificial trachea to assess self-protection properties or the standard air collector to test material properties. The masks were mounted onto the dummy head or standard air collector, and the chamber was closed airtight. After achieving a particle concentration below 35,300 particles/m$^3$ in the chamber, we turned on the syringe pump to reach a steady-state situation inside the chamber with particle counts between $1{,}8 \times 10^7$ to $1{,}8 \times 10^8$ aerosol particles/m$^3$ in each experiment. The sizes of the applied

particles ranged from 0.3 μm to 2 μm what represents the complete diameter range of the generated aerosol. The particle counter was calibrated to an air flow rate of 28.3 L/min and a total measuring range from 0.3 to 10 μm by the manufacturer. The pressurized air was set to 30 L/min to ensure positive pressure and homogeneous particle counts inside the chamber. Pressure equalization with ambient air was ensured by two respirator filters connected in series to the chamber.

The particle concentration was measured alternating by continuously counting either all particles that were penetrating the mask material versus all particles inside the chamber, respectively. The filtration efficacy was calculated by comparing the average particle count in the chamber with the particle count after passing through the mask. We calculated this ratio for particles sized 0.5 μm.

To measure pressure differences of the mask material, the masks were mounted on a standard-sized air collector, the particle-free pressurized airflow was turned on, and the chamber was closed, but no aerosol was admixed. Both the pressurized air and particle counter were run at similar airflow rates.

The statistical analysis of our data was performed using SPSS (IBM, Armonk, USA).

## Results

### Material properties

We assessed the material properties of the masks by measuring both the pressure drop over the mask and average filtration efficacy at a particle size of 0.5 μm. Medical face masks showed the lowest pressure drop, with Type II masks showing lower breathing resistance ($12.9\pm6.8$ Pa/cm$^2$) than non-certified masks ($16.2\pm4.1$ Pa/cm$^2$). Respirators such as the KN95 and FFP2 had two-to-three-fold higher resistances ($32.3\pm7.0$ Pa/cm$^2$ and $26.8 \pm 7.4$ Pa/cm$^2$, respectively), leading to the highest resistance among the tested masks (Fig 2C). The test results for cloth masks varied the most, ranging from 6.9 to 149.3 Pa/cm$^2$.

The average filtration efficacy of the mask materials was the lowest for cloth masks ($27.8\pm25.4\%$), followed by non-EN-certified face masks ($63.4\pm18.7\%$), and the materials of the tested respirators showed very high aerosol retentions (KN95 [$93.8\pm3.9\%$], FFP2 [$98.2\pm1\%$]).

### As-worn filtration efficacy

To determine as-worn filtration efficacy, the masks were mounted on the dummy head and challenged with the test aerosol (Fig 2B). Cloth masks and non-EN-certified face masks had the worst as-worn filtration efficacies in the tested masks, filtering <20% ($11.3 \pm 3.1\%$; $14.2 \pm 2.8\%$) of the 0.5-μm-sized test aerosol fraction. Remarkably, the cloth mask with the highest filtration efficacy in the material test showed the lowest filtration efficacy on the dummy head (84% material filtration efficacy vs. 9% as-worn filtration efficacy), related to a high pressure drop across the mask material (149,3 Pa/cm$^2$) and some masks with low material filtration efficacies showed a comparatively respectable result on the dummy head related to low pressure drop. Hence, there was no direct correlation between the material and dummy head testing.

KN95 respirators performed better than cloth masks and non-certified masks: they reduced the particles by $41.2\% \pm 4\%$. Remarkably, type II medical face masks were not significantly different from KN95 masks, despite having a lower pressure drop. The best overall results were observed for FFP2 masks, with an as-worn filtration efficacy of $65.0 \pm 27\%$ on the dummy head. Notably, face shields did not have any significant ($p< = 0.05$) retention function against the applied aerosols.

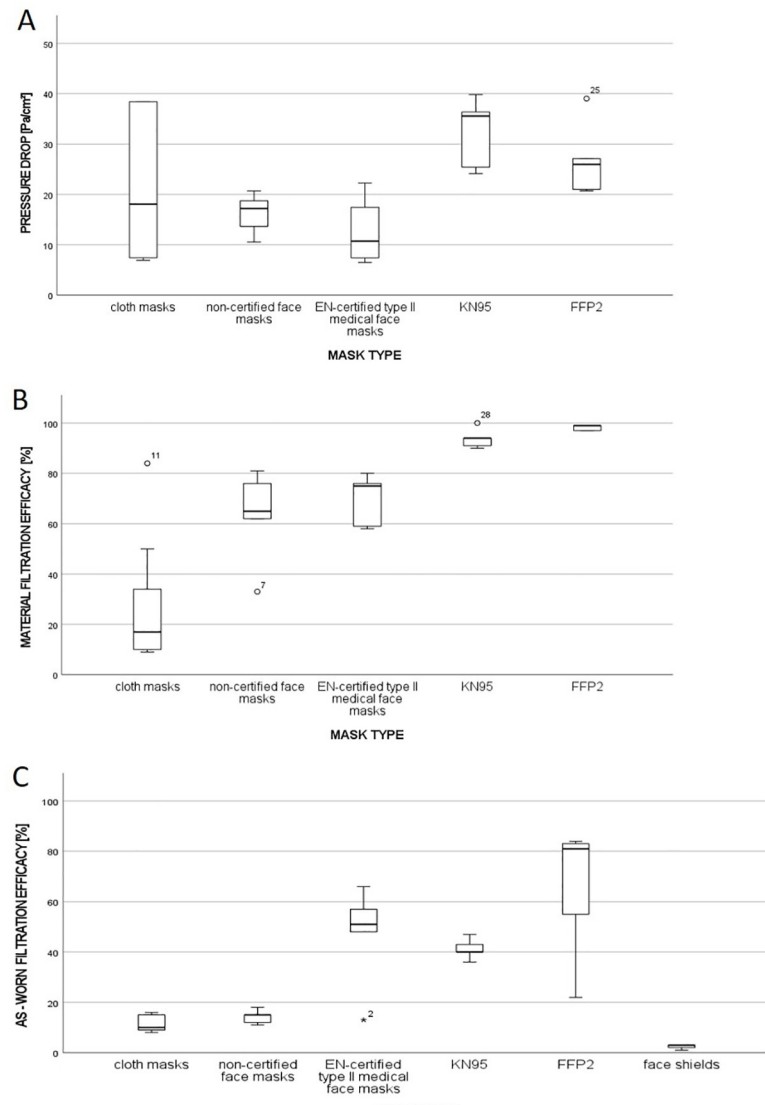

**Fig 2.** A: Distribution of the mean pressure drop across filter fabrics for the mask types tested; B: Distribution of the mean filtration efficacies against aerosol particles sized 0.5 µm for the different tested mask types; C: Distribution of the as-worn filtration efficacies against aerosol particles sized 0.5 µm for the different mask types tested on the dummy head.

On comparing the materials and as-worn filtration efficacies of the masks, four out of five FFP2 and all five KN95 masks reached the upper efficiency range of the tested masks in both the tested parameters, whereas four out of five tested type II medical masks showed as-worn filtration efficacies above average whilst only average material filtration efficacy (Fig 3A). While analyzing the dependence of the as-worn filtration efficacy on the pressure drop, four out of five type II medical face masks and four out of five FFP2 masks showed overall above-average filtration efficacies and low pressure drops and therefore low breathing resistance at the same time (Fig 3B). In contrast, only two KN95 masks reached values above the average for both the parameters. Interestingly, masks with high filtration resistances exhibited the worst values in terms of as-worn filter performance.

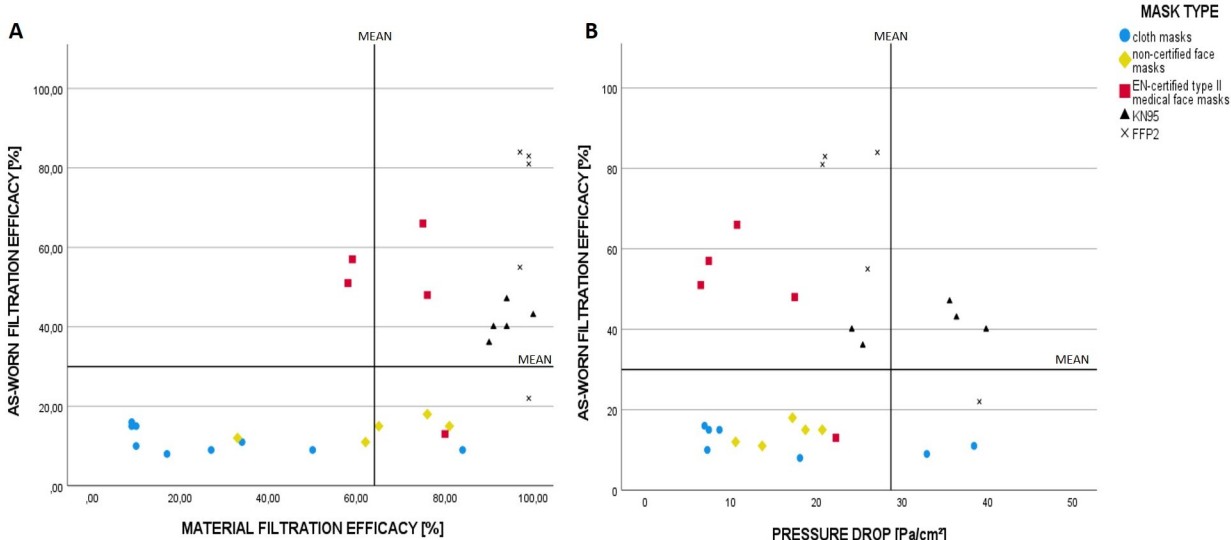

**Fig 3.** As-worn Filtration efficacy in dependence of material filtration efficacy and pressure drop over filter material: A: As-worn filtration efficacies of the mask variants according to the material filtration efficacies; B: As-worn filtration efficacy according to the pressure drops over the filter material of each mask variant. The vertical and horizontal lines represent the mean values of the respective parameters of all masks tested in our study.

The optimal mask effect is a combination of high filter performance and low filter resistance of the material. In our tests, these parameters were achieved by the majority of FFP2 and medical type II face masks (Fig 4). The type II medical masks in our random sample showed very good as-worn filtration performances with a low additional work of breathing at the same time.

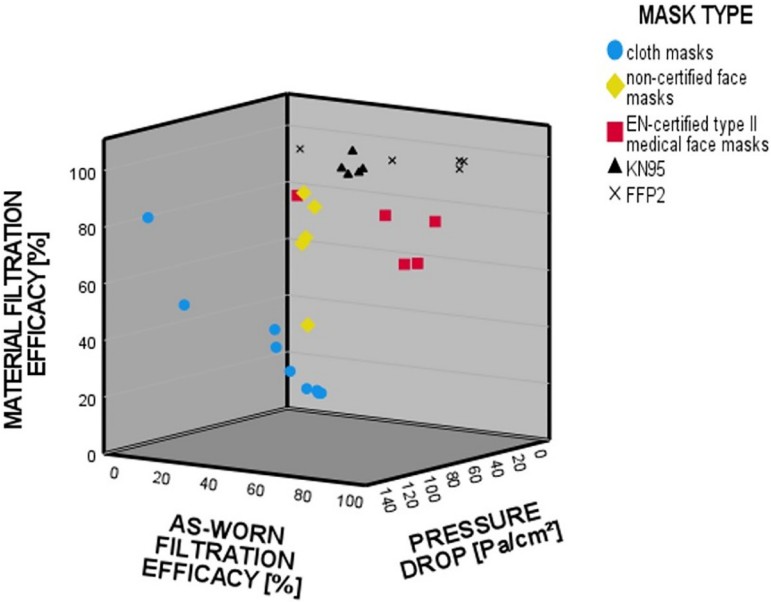

**Fig 4. Interaction between the filter effect of the material and the pressure drop across the material to the total filter performance on the dummy head.**

## Discussion

Our results indicate that it seems recommendable to wear face masks to protect the individual wearer of the mask from inhaling airborne particles. In particular, EN 14683 type II certified medical face masks can provide a high protective effect with low airflow resistance at the same time. Non-certified cloth and medical face masks provided protective effects against our test aerosol; however, these effects were very poor. The cloth masks tested in our study showed high variability between different mask types while differences between masks of the same type were low. This might be due to different used materials, manufacturers and the lack of standardization within this group of masks. Not surprisingly, FFP2 respirators provided the best protective effect on an average. KN95 respirators performed relatively poorly, with filtration efficacies ranging from 36% to 47%. These results are inferior to those of the best medical type II face masks, whose performances ranged from 13% to 66%. The results are remarkable given the higher price, better subjective protection feeling, and higher air flow resistance of the KN95 respirators. However, our findings are consistent with the RAPEX warnings.

Our data also indicates that material filtration efficacy does not necessarily correlate to as-worn filtration efficacy as previously also reported by others [15]. This might be due to the combined effects of mask fit and pressure drop of the mask material and therefore tendency for mask leakage. High pressure drop results in higher breathing resistance and therefore supports leakage, especially if combined to a loosely fitting mask.

Because of the test conditions of the European Norms, EN 14683 and EN 149, the possible protective function of the tested masks in a pandemic situation remains unclear, and some experts have hypothesized that medical face masks only protect others but have no significant self-protecting effect [6]. However, several studies have shown that medical face masks can protect the wearer as much as respirators do [9,16,17]. In contrast to EN 14683, we conducted our experiments using a dummy head with an artificial trachea. Therefore, the mask fit could be assessed for an average head form. The head shape used represents the average shape of American individuals, and presumably, the average head shape in the European area differs from this; nevertheless, according to our knowledge, the corresponding mean values of head shape parameters in Europe were not available at the time of the study. Since a proper fit is crucial for effective protection and because face relief differs among individuals, the filtration efficacy might, however, be different in individuals whose head forms differ significantly from the average. Of note, the applied coating only reflects an approximation of the texture of the human face. However, the use of elastomer coating is cited as a good surrogate for human skin [18]. Given the fact that our dummy head was not as soft as a human head, the tested masks should fit better on a human face than on a dummy.

Our findings were generated using DEHS as a surrogate for coronavirus-containing particles. With this stable aerosol, it is possible to challenge masks with small particles sized ≤0.5 μm. As airborne viruses usually aggregate to form larger particles, testing masks with particles sized 0.5 μm provides a realistic test situation despite SARS-CoV-2 particles being only 60–140 nm in diameter [2,19,20]. If presuming a minute ventilation of 9 L/min and an inspiration to expiration ratio of 1:2, this represents an inspiratory flow rate of 27 L/min. Therefore, the air flow rate used in the experiments is in good approximation to physiological standard parameters of norm ventilation; however, we were unable to demonstrate the consecutive inhalation and exhalation processes with our setup.

Like any virus, SARS-CoV-2 can only infect people as long as it is viable; moreover, a certain number of viable virus particles need to be inhaled to trigger an infection. Thus, the assessed filtration efficacy may differ from the provided protection rate against SARS-CoV-2. Virus-containing particles might dry out during their passage through the mask and lose their

infectivity. Moreover, even a small reduction in inhaled particles might prevent infection or at least lead to a less severe infection [21]. It was shown that the rate of asymptomatic patients was higher among people who wore masks than among those who did not wear masks [21]. The reduction of many, but not all, infective particles might therefore be sufficient to play a key role in infection prevention.

If we consider that the third-party protection properties of masks are approximately as good as the self-protection properties, we can sum up the protective effect when everybody is wearing masks. The basic reproduction number of SARS-CoV-2 ($R_0$) is estimated to vary between 2.24 and 3.58 [22], making it necessary to prevent 55%–72% of the possible transmissions to control the virus. Thus, the control of spread could be achieved in defined environments by either single persons wearing masks with a protective function greater than 55%–72% (e.g., FFP2, N95) or everybody wearing masks with a protection function of 33%–48%. For example, this can be achieved using high-quality medical face masks. Important features of high-quality masks encompass a good fit at the bridge of the nose, low airflow resistance, and EN 14683 conformity. Masks without these features performed in a range comparable to that of cloth masks. However, it is difficult to believe that a protection function of 10%–20% is sufficient for preventing airborne viruses such as SARS-CoV-2 from spreading.

Another argument for medical masks is the fact that respirators induce significantly more discomfort than medical face masks [23]. In our tests, respirators had two-to three-fold higher airflow resistances than medical face masks. This might lead to lower user adherence and consequently to a lower overall protection rate. Therefore, it seems reasonable to widely use medical face masks in hospitals to prevent the virus from spreading, especially if distancing and quarantine are not possible. In situations where a patient cannot wear a mask (e.g., intubation), a medical face mask does not seem sufficient to protect the HCW from SARS-CoV-2. In such cases, respirators such as FFP2 masks should be considered. KN95 respirators should be worn only if other respirators (for example. FFP2/N95) are not available because their filtration efficacy is comparable to that of good medical face masks.

Altogether, our data indicate that it is highly recommendable to wear masks, especially among HCWs, to protect the individual wearer of the mask from an infection with SARS-CoV-2 and other airborne diseases. Medical face masks with good filtration efficacies can provide even better protective effects than KN95 respirators. The assessed filtration efficacy in our experiments presumably underestimates the real infection-preventive effect provided by face masks and respirators. Moreover, the airflow resistances of Medical face masks are lower than that of respirators, leading to higher user adherence and therefore a better overall protection rate. However, further studies need to be conducted to confirm our findings using bioaerosols for example.

## Supporting information

**S1 File.**
(PDF)

## Author Contributions

**Conceptualization:** Frank Günther.

**Data curation:** Christian M. Sterr.

**Formal analysis:** Frank Günther.

**Investigation:** Inga-Lena Nickel, Christina Stranzinger, Claudia I. Nonnenmacher-Winter.

**Methodology:** Christian M. Sterr, Frank Günther.

**Project administration:** Frank Günther.

**Supervision:** Frank Günther.

**Writing – original draft:** Christian M. Sterr.

**Writing – review & editing:** Frank Günther.

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
