## [Decision Letter · Decision Letter 0]

8 Feb 2021

PONE-D-21-01928

Medical face masks offer self-protection against aerosols: An evaluation using a practical in vitro approach on a dummy head

PLOS ONE

Dear Dr. Günther,

Thank you for submitting your manuscript to PLOS ONE. After careful consideration, we feel that it has merit but does not fully meet PLOS ONE’s publication criteria as it currently stands. Therefore, we invite you to submit a revised version of the manuscript that addresses the points raised during the review process.

We look forward to receiving your revised manuscript.

Kind regards,

Amitava Mukherjee, ME, Ph.D.

Academic Editor

PLOS ONE

Journal Requirements:

2. To comply with PLOS ONE submission guidelines, in your Methods section, please provide additional information regarding your statistical analyses. For more information on PLOS ONE's expectations for statistical reporting, please see https://journals.plos.org/plosone/s/submission-guidelines.#loc-statistical-reporting.

3. Please ensure you have discussed any potential limitations of your study in the Discussion.

4. Please ensure that you refer to Figure 1 in your text as, if accepted, production will need this reference to link the reader to the figure.

Reviewers' comments:

Reviewer's Responses to Questions

**Comments to the Author**

1. Is the manuscript technically sound, and do the data support the conclusions?

Reviewer #1: No

Reviewer #2: Yes

Reviewer #3: Yes

2. Has the statistical analysis been performed appropriately and rigorously? 

Reviewer #1: No

Reviewer #2: Yes

Reviewer #3: Yes

3. Have the authors made all data underlying the findings in their manuscript fully available?

Reviewer #1: Yes

Reviewer #2: Yes

Reviewer #3: No

4. Is the manuscript presented in an intelligible fashion and written in standard English?

Reviewer #1: Yes

Reviewer #2: Yes

Reviewer #3: Yes

5. Review Comments to the Author

Reviewer #1: If the authors want to challenge different masks with aerosols and tested the pressure drop, it will be better to use standard testing processes to conduct this study. The references of your standard testing processes should be addressed in the manuscript. Then you can use your own test processes to examine these different masks. Such study methods are more credible and persuasive. Moreover, the word "challenge" is too strong. I suggest you to tone down it.

Reviewer #2: This paper provides an interesting and important analysis of as-worn mask filtration efficiency that is correlated to pressure drop and base material filtration efficiency. I believe some minor edits would improve the readability and impact of the work, and that this manuscript is worthy of publication once these minor edits are addressed. As a colleague, I appreciate your efforts and enjoyed reading your work; best wishes as you continue your endeavors!

Specific comments below; several of these comments are minor or subjective, and do not require addressing before publication in my opinion. Author and editor discretion may be employed when determining when to proceed with publication of this manuscript.

1. The particle size range is listed as 0.3 µm to 2 µm. Does this represent the complete diameter range of the generated aerosol, or is this the limit of the measured range by the instrumentation? If a basic size distribution plot of the base aerosol (i.e. diameter vs. concentration) could be included that would be informative. I understand equipment limitations may make this request more trouble than it is worth.

2. Is there a tolerance available for the measured diameter window? In other words, 0.5 +/- 0.X µm? I’ve found the manufacturers of characterization equipment are reluctant to state this information, so if it is unavailable, this comment may be ignored. What is the sampling rate for the characterization equipment (i.e. 1 sample per second)? Were these averaged for each sample?

3. It may be helpful for cross-comparison to change particle concentration units to SI (#/m3 or, preferably, #/cm3). This suggestion is at the authors’ discretion; I tend to report particulate count data as #/cm3 to mesh with the literature I typically read, but I am aware that other audiences may prefer US Customary units. So if the authors have observed a higher prevalence of particulate concentrations reported as #/ft3 in their target readership, this comment may be ignored.

4. Some may question whether the softness of the dummy head is representative enough of human skin, which will affect fit/leakage, but I applaud the effort here in 3D printing a representative shape and coating with a softer resin. Probably imperfect, but likely as good as any other technique used ad hoc during the pandemic.

5. The variability of filtration efficiency (FE) for cloth mask material (Page 13, line 20) is very high- that may be accurate (I’ve observed somewhat lower but similar variability) but calling for a double-check in case there’s a mistake. Was this variability due to one sample providing much higher FE than average, or much lower? Or was variability just high throughout?

6. The finding that base material FE does not necessarily translate to as-worn FE (i.e. material that filters well in ideal testing doesn’t always make the best mask) is interesting and important. The correlation of filtration efficiency and breathability (pressure drop) is useful and instructive? This finding has been hinted at in earlier work (Hill et al, doi:10.1021/acs.nanolett.0c03182) but is better represented here. Figure 4 is a little hard to interpret, however (image quality is a little low, too, which doesn’t help). Does “overall filtration efficiency” mean the FE recorded when worn by the headform, whereas “material filtration efficiency” corresponds to the sealed measurement of base material FE alone? If so, perhaps consider rephrasing for clarity (maybe “as-worn mask filtration efficiency” or something similar. Also, on Figure 4 (right), would it be useful to plot the inverse pressure drop so that highest breathability & FE appear in the top right quadrant? It’s a minor adjustment that might better communicate the trend.

7. Formatting adjustments to the charts would improve readability- suggest eliminating decimal places on the axes, increasing font size throughout, and capitalizing axis titles & legend labels.

8. The discussion of spread control as related to mask usage and filtration efficiencies (page 17) is fascinating. I lack the epidemiological expertise to adequately evaluate the claims made here but find this an interesting approach.

9. A better reference for the size of the SARS-CoV-2 virion might be the original source, which I believe to be: Zhu, N. et al, A Novel Coronavirus from Patients with Pneumonia in China, 2019. N. Engl. J. Med. 2020, 382, 727– 733, DOI: 10.1056/NEJMoa2001017

The following are a few minor editing notes (suggestions to improve readability, typos to correct, etc.):

Page 11, line 8: Move semicolon from after “masks” to after “defects” to improve readability

Page 13 line 12: Suggest changing “better” to a more neutral term (i.e. “lower” or “more breathable”, though I suppose the term “breathable” may not be empirically accurate in this context)

Page 13 Line 20: Typo in the non-EN-certified face mask efficiency (63.4.0)

Page 14 line 13: I understand what you mean by saying “slightly better results than KN95 masks (not significant”, but it may be considered more accurate to simply say “were not significantly different from KN95 masks”. May be a semantic issue; suggest adjusting if you agree.

Reviewer #3: This manuscript studied if it is advised to use face masks to protect the human wearer by comparing respirators, medical face masks, and cloth masks. The filtration efficacy and breathing resistance of the mask material were determined. The overall filtration efficiency of simulating real live conditions on a dummy head was also compared. The topic is interesting and urgent under the COVID-19 pandemic. However, a few questions should be addressed and the reviewer would appreciate further discussion in the manuscript. The reviewer recommends a major revision of this manuscript before its publication.

General comments:

1. Lack of information

1) The authors stated that 32 different face coverings were tested. Please list details of the materials information in the Table (e.g., name, brand, materials, etc.), as the results are likely to vary widely among the various brands or models.

2) Page 11, Methods section, Line 11, the fit factor was mentioned in this sentence but did not introduce throughout the paper.

2. Reference

1) Page 11, Methods section, Line 15, some references are necessary to sustain the skin-like surface and the techniques of coated with liquid gum.

2) Page 16, Discussion, Line 16-18, “The chosen airflow rate of 28.3 L/min in our experiments provides a realistic test condition considering a minute volume of 18 L/min and an inspiration/expiration ratio of 1:2”, some related references should be taken into account to support the selected data.

3. Figures

1) Please improve the quality of the figures.

2) Page 14, Overall filtration efficiency section, Line 22-25, as mentioned “figure 3A and figure 3B” here, but the following figure 3 does not contain “A” and “B”.

3) Page 15, Overall filtration efficiency section, Line 4-8, figure 4 is good and essential to indicate the tested results to be seen, but only a few explanations are given in figure 4. The authors also need to illustrate the meaning of lines in this figure.

Detailed comments:

1. Page 13, Material properties section, Line 19, the authors showed the efficiency of cloth masks and non-EN-certified face masks. Please clarify what these efficiencies are: average efficiency or specific efficiency among the tested materials?

2. Page 14, Overall filtration efficiency section, Line 6-10, why are the results of the material test and dummy head testing not matched? Any potential explanations should be suggested by the authors.

3. Page 16, Discussion, Line 14, “With this stable aerosol, it is possible to challenge masks with small particles sized ≤0.5 µm”, why the particles sizes below 0.5 µm are not taken into consideration in the paper since the aerosols can be generated? In addition, SARA-CoV-2 particles are only 60-140 nm in diameter.

6. PLOS authors have the option to publish the peer review history of their article (what does this mean?). If published, this will include your full peer review and any attached files.

Reviewer #1: No

Reviewer #2: **Yes: **W. Cary Hill

Reviewer #3: No

---

## [Author Response · Author response to Decision Letter 0]

16 Feb 2021

Dear Dr. Mukherjee

First of all, we would like to thank you for giving us the opportunity to revise the manuscript. The revision includes the following changes according to the reviewers’ comments. In the following section, we have responded to all the comments in detail and have explained the changes made to the manuscript (highlighted in yellow). We also would like to thank the reviewers for their attentive and thorough review and for their useful comments. We feel that the manuscript has been improved due to their work.

Ref.: PONE-D-21-01928

Reviewer #1: 

If the authors want to challenge different masks with aerosols and tested the pressure drop, it will be better to use standard testing processes to conduct this study. The references of your standard testing processes should be addressed in the manuscript. Then you can use your own test processes to examine these different masks. Such study methods are more credible and persuasive. Moreover, the word "challenge" is too strong. I suggest you to tone down it.

Response: We thank the reviewer for the suggestion. With exception of cloth masks and non-certified face masks, test certificates based on standard methods are available for the masks tested in our study. Re-testing or comparison of the masks using standard procedures was not within the scope of our study. Rather, the aim was to create a more realistic and meaningful test environment for the self-protection function of various mask types against aerosols.

Reviewer #2: 

This paper provides an interesting and important analysis of as-worn mask filtration efficiency that is correlated to pressure drop and base material filtration efficiency. I believe some minor edits would improve the readability and impact of the work, and that this manuscript is worthy of publication once these minor edits are addressed. As a colleague, I appreciate your efforts and enjoyed reading your work; best wishes as you continue your endeavors!

Specific comments below; several of these comments are minor or subjective, and do not require addressing before publication in my opinion. Author and editor discretion may be employed when determining when to proceed with publication of this manuscript.

1. The particle size range is listed as 0.3 µm to 2 µm. Does this represent the complete diameter range of the generated aerosol, or is this the limit of the measured range by the instrumentation? If a basic size distribution plot of the base aerosol (i.e. diameter vs. concentration) could be included that would be informative. I understand equipment limitations may make this request more trouble than it is worth.

Response: The listed particle range represents the size range of the generated aerosol. The measuring range of the applied instrumentation is 0.3-10 µm. The application of a complex aerosol mixture in our experiments was therefore adapted to the real life situation. We have clarified that and added the missing information in the manuscript.

2. Is there a tolerance available for the measured diameter window? In other words, 0.5 +/- 0.X µm? I’ve found the manufacturers of characterization equipment are reluctant to state this information, so if it is unavailable, this comment may be ignored. What is the sampling rate for the characterization equipment (i.e. 1 sample per second)? Were these averaged for each sample?

Response: Unfortunately, there is no statement available on the measurement deviation of the used device. However, it was calibrated by using defined test particles of the size range that was also applied in our experiments and therefore delivered reliable and discriminative results. Sampling took place continuously within defined measurement intervals by detecting absolute numbers of particles. We have addressed this now in the text.

3. It may be helpful for cross-comparison to change particle concentration units to SI (#/m3 or, preferably, #/cm3). This suggestion is at the authors’ discretion; I tend to report particulate count data as #/cm3 to mesh with the literature I typically read, but I am aware that other audiences may prefer US Customary units. So if the authors have observed a higher prevalence of particulate concentrations reported as #/ft3 in their target readership, this comment may be ignored.

Response: We have changed that accordingly.

4. Some may question whether the softness of the dummy head is representative enough of human skin, which will affect fit/leakage, but I applaud the effort here in 3D printing a representative shape and coating with a softer resin. Probably imperfect, but likely as good as any other technique used ad hoc during the pandemic.

Response: We agree with the reviewer. The applied coating only reflects an approximation of the texture of the human face. However, the use of elastomer coating is cited as a good surrogate in various publications. We have now provided an additional reference for skin like coatings.

5. The variability of filtration efficiency (FE) for cloth mask material (Page 13, line 20) is very high- that may be accurate (I’ve observed somewhat lower but similar variability) but calling for a double-check in case there’s a mistake. Was this variability due to one sample providing much higher FE than average, or much lower? Or was variability just high throughout?

Response: We agree with the reviewer. The cloth masks tested in our study showed high variability between different mask types while differences between masks of the same type were low. This might be due to different used materials, manufacturers and the lack of standardization within this group of masks. We have addressed this now in the text.

6. The finding that base material FE does not necessarily translate to as-worn FE (i.e. material that filters well in ideal testing doesn’t always make the best mask) is interesting and important. The correlation of filtration efficiency and breathability (pressure drop) is useful and instructive? This finding has been hinted at in earlier work (Hill et al, doi:10.1021/acs.nanolett.0c03182) but is better represented here. Figure 4 is a little hard to interpret, however (image quality is a little low, too, which doesn’t help). Does “overall filtration efficiency” mean the FE recorded when worn by the headform, whereas “material filtration efficiency” corresponds to the sealed measurement of base material FE alone? If so, perhaps consider rephrasing for clarity (maybe “as-worn mask filtration efficiency” or something similar. Also, on Figure 4 (right), would it be useful to plot the inverse pressure drop so that highest breathability & FE appear in the top right quadrant? It’s a minor adjustment that might better communicate the trend.

Response: We thank the reviewer for the helpful comment. We have added a paragraph on this subject to the discussion. Furthermore, overall filtration efficacy was changed to as-worn filtration efficacy throughout the manuscript. 

7. Formatting adjustments to the charts would improve readability- suggest eliminating decimal places on the axes, increasing font size throughout, and capitalizing axis titles & legend labels.

Response: We have revised the figures, axis titles and legend labels.

8. The discussion of spread control as related to mask usage and filtration efficiencies (page 17) is fascinating. I lack the epidemiological expertise to adequately evaluate the claims made here but find this an interesting approach.

Response: We thank the reviewer for the comment.

9. A better reference for the size of the SARS-CoV-2 virion might be the original source, which I believe to be: Zhu, N. et al, A Novel Coronavirus from Patients with Pneumonia in China, 2019. N. Engl. J. Med. 2020, 382, 727– 733, DOI: 10.1056/NEJMoa2001017

Response: We thank the reviewer for the comment and have added the mentioned reference for the size of the SARS-CoV-2 virion.

The following are a few minor editing notes (suggestions to improve readability, typos to correct, etc.):

Page 11, line 8: Move semicolon from after “masks” to after “defects” to improve readability

Response: We have corrected that accordingly.

Page 13 line 12: Suggest changing “better” to a more neutral term (i.e. “lower” or “more breathable”, though I suppose the term “breathable” may not be empirically accurate in this context)

Response: We have changed that accordingly.

Page 13 Line 20: Typo in the non-EN-certified face mask efficiency (63.4.0)

Response: We have corrected that accordingly.

Page 14 line 13: I understand what you mean by saying “slightly better results than KN95 masks (not significant”, but it may be considered more accurate to simply say “were not significantly different from KN95 masks”. May be a semantic issue; suggest adjusting if you agree.

Response: We agree with the reviewer and have changed that accordingly.

Reviewer #3: 

This manuscript studied if it is advised to use face masks to protect the human wearer by comparing respirators, medical face masks, and cloth masks. The filtration efficacy and breathing resistance of the mask material were determined. The overall filtration efficiency of simulating real live conditions on a dummy head was also compared. The topic is interesting and urgent under the COVID-19 pandemic. However, a few questions should be addressed and the reviewer would appreciate further discussion in the manuscript. The reviewer recommends a major revision of this manuscript before its publication.

General comments:

1. Lack of information

1) The authors stated that 32 different face coverings were tested. Please list details of the materials information in the Table (e.g., name, brand, materials, etc.), as the results are likely to vary widely among the various brands or models.

Response: We agree with the reviewer and have now provided the available information on the tested masks as Supplement 1. 

2) Page 11, Methods section, Line 11, the fit factor was mentioned in this sentence but did not introduce throughout the paper.

Response: We apologize for the confusion. The term “fit factor” is not applicable in that mentioned context, since we did not calculate a fit factor in our study. We therefore replaced “fit factor” by “mask fit” throughout the manuscript.

2. Reference

1) Page 11, Methods section, Line 15, some references are necessary to sustain the skin-like surface and the techniques of coated with liquid gum.

Response: We have now provided the reference. (see also: response to reviewer 1 comment 4)

2) Page 16, Discussion, Line 16-18, “The chosen airflow rate of 28.3 L/min in our experiments provides a realistic test condition considering a minute volume of 18 L/min and an inspiration/expiration ratio of 1:2”, some related references should be taken into account to support the selected data.

Response: If presuming a minute ventilation of 9 L / min and an inspiration to expiration ratio of 1 : 2, this represents an inspiratory flow rate of 27 L / min respectively (inspiration of 9L in 20s). Therefore, a flow rate of 28.3 L/min is in good approximation to physiological standard parameters during norm ventilation. We have now explained this in more detail.

3. Figures

1) Please improve the quality of the figures.

Response: We have revised the figures, axis titles and legend labels.

2) Page 14, Overall filtration efficiency section, Line 22-25, as mentioned “figure 3A and figure 3B” here, but the following figure 3 does not contain “A” and “B”.

Response: We apologize for the confusion. We have corrected the accordingly.

3) Page 15, Overall filtration efficiency section, Line 4-8, figure 4 is good and essential to indicate the tested results to be seen, but only a few explanations are given in figure 4. The authors also need to illustrate the meaning of lines in this figure.

Response: We have renamed the figure and have added additional labeling in Figure 3.

Detailed comments:

1. Page 13, Material properties section, Line 19, the authors showed the efficiency of cloth masks and non-EN-certified face masks. Please clarify what these efficiencies are: average efficiency or specific efficiency among the tested materials?

Response: We have clarified the statement.

2. Page 14, Overall filtration efficiency section, Line 6-10, why are the results of the material test and dummy head testing not matched? Any potential explanations should be suggested by the authors. 

Response: We have added additional information on the influence of the pressure drop across the mask material to this section.

3. Page 16, Discussion, Line 14, “With this stable aerosol, it is possible to challenge masks with small particles sized ≤0.5 µm”, why the particles sizes below 0.5 µm are not taken into consideration in the paper since the aerosols can be generated? In addition, SARA-CoV-2 particles are only 60-140 nm in diameter.

Response: The common carrier–to-virus particle size ratio for stable and infective aerosols is approximately 1/10. Therefore, a smallest particle size range between 0.5 to 0.6 µm would play a key role in transmission of SARS-CoV-2. For this reason, we focused on 0.5 µm as the particle size to be investigated in our study. An according reference was cited in the manuscript (Agranovski et. al. (18)). 

Our present revision addresses all points made by the reviewers. We replied point-by-point to their comments and highlighted all changes to the manuscript as suggested.

We hope that the manuscript is now eligible for publication in your journal.

Yours sincerely,

Frank Günther

---

## [Decision Letter · Decision Letter 1]

22 Feb 2021

Medical face masks offer self-protection against aerosols: An evaluation using a practical in vitro approach on a dummy head

PONE-D-21-01928R1

Dear Dr. Günther,

We’re pleased to inform you that your manuscript has been judged scientifically suitable for publication and will be formally accepted for publication once it meets all outstanding technical requirements.

Kind regards,

Amitava Mukherjee, ME, Ph.D.

Academic Editor

PLOS ONE

Additional Editor Comments (optional):

Reviewers' comments:

Reviewer's Responses to Questions

**Comments to the Author**

1. If the authors have adequately addressed your comments raised in a previous round of review and you feel that this manuscript is now acceptable for publication, you may indicate that here to bypass the “Comments to the Author” section, enter your conflict of interest statement in the “Confidential to Editor” section, and submit your "Accept" recommendation.

Reviewer #2: All comments have been addressed

Reviewer #3: All comments have been addressed

2. Is the manuscript technically sound, and do the data support the conclusions?

Reviewer #2: Yes

Reviewer #3: Yes

3. Has the statistical analysis been performed appropriately and rigorously? 

Reviewer #2: Yes

Reviewer #3: Yes

4. Have the authors made all data underlying the findings in their manuscript fully available?

Reviewer #2: Yes

Reviewer #3: Yes

5. Is the manuscript presented in an intelligible fashion and written in standard English?

Reviewer #2: Yes

Reviewer #3: Yes

6. Review Comments to the Author

Reviewer #2: I appreciate the improvements made to the manuscript and believe it is ready for publication- nicely done! Last comment would be to remove the extraneous decimal places in Figure 3A. Otherwise, it is good to go.

Reviewer #3: (No Response)

7. PLOS authors have the option to publish the peer review history of their article (what does this mean?). If published, this will include your full peer review and any attached files.

Reviewer #2: **Yes: **W. Cary Hill

Reviewer #3: No

---

## [Editor Report · Acceptance letter]

25 Feb 2021

PONE-D-21-01928R1 

Medical face masks offer self-protection against aerosols: An evaluation using a practical in vitro approach on a dummy head 

Dear Dr. Günther:

I'm pleased to inform you that your manuscript has been deemed suitable for publication in PLOS ONE. Congratulations! Your manuscript is now with our production department. 

Kind regards, 

on behalf of

Professor Dr. Amitava Mukherjee 

Academic Editor

PLOS ONE